# The Role of Surgery in Lung Cancer Treatment: Present Indications and Future Perspectives—State of the Art

**DOI:** 10.3390/cancers13153711

**Published:** 2021-07-23

**Authors:** François Montagne, Florian Guisier, Nicolas Venissac, Jean-Marc Baste

**Affiliations:** 1Department of Thoracic Surgery, Calmette Hospital, University Hospital of Lille, Boulevard du Pr. J Leclercq, F-59000 Lille, France; francois.montagne@chru-lille.fr (F.M.); nicolas.venissac@chru-lille.fr (N.V.); 2Department of Pneumology, Rouen University Hospital, 1 rue de Germont, F-76000 Rouen, France; florian.guisier@chu-rouen.fr; 3Clinical Investigation Center, Rouen University Hospital, CIC INSERM 1404, 1 rue de Germont, F-76000 Rouen, France; 4Faculty of Medicine and Pharmacy of Rouen, Normandie University, LITIS QuantIF EA4108, 22 Boulevard Gambetta, F-76183 Rouen, France; 5Department of General and Thoracic Surgery, Rouen University Hospital, 1 rue de Germont, F-76000 Rouen, France; 6Faculty of Medicine and Pharmacy of Rouen (UNIROUEN), Normandie University, INSERM U1096, 22 Boulevard Gambetta, F-76000 Rouen, France

**Keywords:** lung cancer, minimally invasive surgery, video-assisted thoracoscopic surgery, robotic-assisted thoracoscopic surgery, enhanced postsurgical recovery, surgical diagnosis, palliative supportive care

## Abstract

**Simple Summary:**

Lung cancer evolutions, innovative systemic treatments, minimally invasive thoracic surgery approaches and perioperative medical care have changed the role of surgery in the treatment of lung cancer. Pre-invasive and early-stage lung cancer, and conversely, advanced and metastatic tumors can be treated by innovative imaging-guided resection, minimally invasive approach or hybrid approach with very good short-term outcomes, enhanced recovery and preserved long-term survival. Considering lung cancer as a chronic disease, surgery must anticipate future disease evolution by sparing lung tissue and preserving lung function, while an oncologic complete resection must be performed. Surgery could also be valuable when recurrences occur or for selected palliative conditions. This article outlines present indications and future perspectives of lung surgery in lung cancer.

**Abstract:**

Non-small cell lung cancers (NSCLC) are different today, due to the increased use of screening programs and of innovative systemic therapies, leading to the diagnosis of earlier and pre-invasive tumors, and of more advanced and controlled metastatic tumors. Surgery for NSCLC remains the cornerstone treatment when it can be performed. The role of surgery and surgeons has also evolved because surgeons not only perform the initial curative lung cancer resection but they also accompany and follow-up patients from pre-operative rehabilitation, to treatment for recurrences. Surgery is personalized, according to cancer characteristics, including cancer extensions, from pre-invasive and local tumors to locally advanced, metastatic disease, or residual disease after medical treatment, anticipating recurrences, and patients’ characteristics. Surgical management is constantly evolving to offer the best oncologic resection adapted to each NSCLC stage. Today, NSCLC can be considered as a chronic disease and surgery is a valuable tool for the diagnosis and treatment of recurrences, and in palliative conditions to relieve dyspnea and improve patients’ comfort.

## 1. Introduction

Lung cancer is the leading cause of cancer mortality worldwide with 2.09 million newly diagnosed cases in 2018 and 1.76 million people related deaths [1,2]. Lung cancer is “a generic term” of a very heterogenous disease by its histology, its molecular intrinsic characteristics and variabilities, its clinical stages at diagnosis, and patient conditions. This heterogeneity determines which treatment, combinations of treatment, or multimodality therapy (MMT) plan can be proposed in a complete treatment pathway integrating surgery.

Most lung cancers, approximately 70%, are diagnosed at an advanced stage [3], but the increasing use of screening programs and other follow-up have led to the earlier diagnosis of less-advanced stages, reducing lung cancer mortality [4,5]. These earlier and potentially resectable tumors raise questions regarding lung sparing surgery, minimally invasive approaches, the type of lymph node dissection (LND), and the best surgical treatment for patients with low respiratory capacity or elderly patients, compared to non-ablative local therapy. In addition, thanks to prolonged survival after curative treatments, lung cancer can be considered as a chronic disease and recurrences need to be anticipated with the aim of treating them with local therapies, such as surgery and radiotherapy, associated with systemic therapy.

The main aim of oncologic surgery in lung cancer is to cure lung cancer using a personalized surgical resection in a complete treatment pathway. Other aims are to contribute to the diagnosis of recurrences, to provide treatments, and to relieve symptoms in palliative conditions while continuing to improve patients’ quality of life (QoL). Today, surgeons are involved in the many steps of the lives of patients affected by lung cancer, from curative to palliative treatments.

In this article, we will review some aspects of lung cancer surgery for non-small cell lung cancer (NSCLC), its present indications, and future perspectives according to the outline presented above.

## 2. History—Medical and Surgical Evolutions of Lung Cancer

### 2.1. Lung Cancer Epidemiology—Past and Recent Trends

Cancer incidence and mortality are increasing worldwide due to aging, growth of the population, and changes in prevalence of the main risk factors, associated with socioeconomic development. In 2018, lung cancer was the first diagnosed cancer and the leading cause of cancer-related death despite an overall increased survival [2]. Lung cancer incidence and mortality are closely linked to cigarette smoking patterns. Following tobacco prevention programs in industrialized countries, lung cancer incidence and mortality have decreased, and the same trends are expected in emerging economies and developing countries [1]. Nevertheless, socioeconomic and educational inequalities, as well as diagnosis at later stages of disease, contribute to variability in lung cancer incidence and mortality even in industrialized countries [6], with an average 5-year survival of 13% in Europe [7] and of 19% in the United States [3]. In emerging economies, tobacco use is a major barrier to effective cancer control, but other important risk factors have been described including environmental pollution, and workplace exposure [8]. Concerning developing countries, due to the lack of reliable registries, lung cancer incidence and mortality rates seem to be low. Other explanations have been proposed as differences in trends in the tobacco epidemic, a lower life expectancy of the population, and less access to health care [1].

### 2.2. Lung Cancer Screening Programs

More early-stage NSCLC is being detected, thanks to screening and early detection programs for lung cancer in high-risk populations [4,5]. In addition, more early-stage NSCLC is being identified on CT scan (Figure 1) for patients followed-up for other cancers, or for thoracic symptoms, or for COVID-19 symptoms.

Screening programs have shown a reduction in lung cancer mortality rates of 20% and 24%, respectively, in the U.S.-based National Lung Screening Trial [4] (NLST) and in the Dutch-Belgian lung-cancer screening randomized trial [5]. Treating early or pre-invasive NSCLC, in younger and non-symptomatic patients raises other questions such as performing oncologic sublobar resection, in order to spare lung function because these patients have a high risk of developing another lung cancer that could also be treated by surgery.

### 2.3. Developments in “Medicine”, Re-Thinking Care

Medicine has evolved a lot through the ages, and Hippocrates’ aphorism “Cure sometimes, treat often, comfort always” has taken on even more meaning because we take care of our patients in a global way, because we have the possibility to do so, with the objective of improving not only life, but also the QoL.

Surgical lung tumor resection is “not only a step” in a patient’s personalized curative treatment, but also the opportunity to change some of their bad lifestyles such as smoking, drinking, sedentary lifestyle, and unbalanced diet. A surgical intervention represents a particular event in a patient’s life, which physically impacts the body with its own emotional symbolism, and more than medical treatments as chemotherapy, immunotherapy, and radiotherapy, surgery could provide the opportunity to improve the patient’s overall health condition. Our responsibility as physicians is to do everything possible to provide the best care for our patients. Surgeons must do more than just performing the best oncologic resection, with the best short and long-term outcomes, they must also promote good health, by encouraging patients to stop their bad lifestyle habits, and to start pre-operative functional rehabilitation in order to improve functional outcomes leading to reduced post-operative complications.

#### 2.3.1. Enhanced Recovery after Surgery and Prehabilitation

The concept of Enhanced Recovery After Surgery (ERAS) was first defined in 1997 by Professor Henrik Kehlet for colorectal surgery [9]. It was initially based on 6 pillars: preoperative information and education, attenuation of stress, pain relief, exercise, enteral nutrition, and growth factors. The objective of ERAS is to enhance recovery after surgery by reducing morbidity and post-operative complications, with a better surgical experience [10,11,12]. Today it has been widely extended to other specialties. In thoracic surgery, ERAS preoperative protocols combine educational programs, treatments and prevention of undernutrition, tobacco and alcohol addictions, and optimization of patients’ respiratory and cardiac comorbidities. Physiotherapy programs are also important. ERAS perioperative protocols are based on a shorter fasting period, sedative drug avoidance, opioid sparing, minimally invasive surgical approaches, with fast removal of chest tubes and catheters in order to enhance self-mobilization, to reduce post-operative complications and adverse events, and to improve post-operative functional state [10,11,12]. Surgeons have a determining role in patient information and education to promote cohesion to ERAS. 

As an echo to the ERAS concept, respiratory prehabilitation has recently shown its positive impact in the management of respiratory fragile patients by reducing short-term complications and adverse events [13,14] as well as improving the surgical experience. Nevertheless, these programs must be short or started very early at the time of diagnosis, in order not to delay therapeutic management to prevent cancer progression.

#### 2.3.2. Quality of Life Assessments—“Surgery for a Better Life”

Disease, lung cancer, comorbidities, lung resection, systemic therapy and radiotherapy have a detrimental impact on a patient’s QoL which has recently gained much attention [15,16,17,18]. QoL reflects multi-dimensional items [17] as social, physical, and mental well-being, as well as functional status and provides further information usually reported as morbidity and mortality with short-term outcomes. Due to the diversity of QoL scales, specific or global, surgical and medical results could be debated [19,20]. 

Combining all progress made in lung cancer screening programs, minimally invasive approaches, anesthesia and preoperative rehabilitation, systemic antineoplastic drugs, and radiotherapy, lung cancer can now be managed as a chronic disease and QoL should be a new criterion of interest for any new therapeutic assessment with a standardized QoL scale in order to improve the reproducibility of this field of research. Living better is just as important as living longer.

### 2.4. History of Lung Cancer Surgery, from Past to Current Trends—“Smaller Is Better”

“Smaller resections, smaller incisions and better functional outcomes”.

Surgical resection for cancer is the first anti-neoplastic treatment which was developed a century ago but lung cancer surgery is still young. At the beginning, lung cancer resection was considered as “a simple and easy resection” compared to a resection performed for tuberculosis but it was rarely performed because diagnoses were made at advanced stages.

The first systematic description of lung resection for cancer was made by Graham and Singer 90 years ago. They performed a successful pneumonectomy, which became the gold standard procedure for lung cancer [21,22] but was not routinely performed until the 50′s. In 1950, Churchill et al. [23] presented strong results in favor of lung lobectomy compared to pneumonectomy and 2 years later, lobectomy was considered as a valid oncologic resection for lung cancer. Its indication switched from “a lobectomy because a pneumonectomy is contraindicated” to “a lobectomy should be performed if it can be done” and today lobectomy with complete LND is still the gold standard lung cancer resection [24,25,26,27,28,29]. Today, sublobar resection is discussed in order to spare lung tissue and function for early-stage tumors, and not only for compromised patients.

Concerning the surgical approach, thoracotomy remains the gold standard. Minimally invasive surgery (Figure 2), as video-assisted thoracoscopic surgery (VATS) and robotic-assisted thoracoscopic surgery (RATS) constitutes a real technological, medical and surgical revolution by making it possible to carry out a resection that respects oncologic criteria, using “small incisions” and without rib spreading.

Today, VATS and RATS are indicated for the resection of an early-stage NSCLC, clinical stage I [28,29], because their efficacy and safety have been proved. Compared to thoracotomy, a VATS lung resection [30,31,32,33,34,35,36] in two randomized controlled trials [33,37] or a RATS lung resection [38,39,40,41,42,43] led to better short-term outcomes with fewer adverse events, shorter hospital stays, and lower morbidity and mortality rates. Regarding short-term outcomes the superiority of VATS or RATS is still debated [38,39,42]. Concerning the long-term outcomes of VATS and RATS, overall survival (OS) and disease-free survival (DFS) are major criteria of oncologic quality used to evaluate the resection performed for any cancer. Results are debated concerning operative lymph node staging, and nodal upstaging in open surgery, VATS or RATS [44,45,46,47,48,49,50,51,52] but more than leading to enhanced recovery, VATS and RATS lead to preserve long-term survival [34,38,48,53,54,55,56,57,58].

During the last decades, thanks to the progress of anesthesia and surgery, lung cancer surgery has gone from being a potentially life-threatening surgery to one of the safest surgeries as illustrated by the decrease in postoperative mortality. During the 60′s, the post-operative mortality rate was 17% after pneumonectomy, 10% after lobectomy [59], and 9% after simple exploratory thoracotomy, when a resection could not be performed. Then, post-operative mortality decreased significantly, but with considerable variations according to the series and databases. During the 2000s, in-hospital mortality rate decreased to 8% after pneumonectomy and 3% after lobectomy [60]. Today, the 30-day mortality rate has further decreased to 2% after open lobectomy, 1.3% after minimally invasive lobectomy (61), and 4% after pneumonectomy. At the same time, lung resection morbidity has also decreased, as well as the length of hospital stay to the point of being able to perform ambulatory lobectomy [61,62,63].

Because the aim of surgery is to perform “R0 resection” with the best lymph node assessment, complete LND is recommended [29,64,65,66,67] because it is the guarantor of accurate lymph node staging and is not associated with more complications compared to other lymph node assessment [68,69,70,71]. The lobe specific lymph node assessment, defined in surgical guidelines [66,67], could be performed for specific cases as cT1 tumors [66], or tumors of less than 30 mm without metastasis in the first lobar node, N1 area, on frozen section [67], and seems to lead to a lower up-staging rate, without any adverse outcomes concerning long-term survival [72,73,74,75].

### 2.5. Operating on Our Elders—A Switch to Radiotherapy?

The average age for diagnosis of NSCLC is 70 years in industrialized countries and NSCLC is therefore a disease of the elderly. Today, due to increasing life expectancy, and the opportunity to age better and better, elderly patients are in good physical shape for longer. 

To treat elderly patients, we need to take into account a patient’s life expectancy and preferences, functional age, comorbidities and estimated benefits and risks [76]. Latest guidelines from the EORTC Task Force Lung Cancer Group and International Society for Geriatric Oncology published in 2014 [76] are helpful to better diagnose and treat elderly patients. But since 2014, many changes have occurred in the medical and surgical treatment of NSCLC. Due to increased frailty and operative risks in elderly patients, segmentectomy is discussed as an alternative to lobectomy in early-stage NSCLC [77], allowing a precise diagnosis associated with LND with no increase in the morbidity-mortality of this procedure [78,79] and treatment or a shift to stereotaxic body radiotherapy (SBRT) with good results. SBRT is the prevailing treatment in octogenarians for stage I NSCLC, more than 80%, and its wider application finally reflects the frailty diversity of these patients. Nevertheless, results are debated, because surgery compared to SBRT enhanced OS after 2 years from the treatment [80] and other trials have failed to give strong results [81]. In modern surgical management of early-stage lung cancer with personalized resection, using a minimally invasive approach allowing very low morbidity and no mortality, within the framework of ERAS protocols, surgery has 2 major advantages compared to radiotherapy. First, it allows a definitive histopathological diagnosis to be made and acute staging including lymph node assessment and second, it allows a controlled resection of the tumor, preserving QoL.

## 3. Lung Cancer and Surgical Treatment

### 3.1. Surgical Treatment for Cancer Patients Today

In each lung cancer condition, whether pre-invasive, early stage, advanced stage, after full medical treatment or “salvage surgery”, the challenges of surgery are to be oncologic and anatomical, ideally without morbidity and mortality, with the least functional impact and the best QoL, preserving long-term survival, and anticipating treatment options for recurrences. The ways to achieve this are specific to the stage of lung cancer and tailored to the patient.

Thus, for example, minimally invasive approaches, sublobar resection and LND are indicated for early-stage NSCLC [28,29], while for advanced-stage and metastatic lung cancer, surgery has been redefined in an MMT plan, including targeted therapy and immunotherapy [82], as well as lobectomy, or bilobectomy or pneumonectomy with LND by thoracotomy. 

For each case, it is essential to discuss in multidisciplinary meeting all the different therapeutic options in order to propose to the patient the most appropriate treatment, innovative therapeutics, inclusion in trials, to allow him/her prolonged survival and the best QoL.

### 3.2. Surgical Treatment for Early Stages—Stage I NSCLC

#### 3.2.1. Surgical Options

Open lobectomy with complete LND is the gold standard surgical resection for stage I NSCLC (Figure 3) [28,29,65,66,67] but alternatives concerning the extent of resection, the extent of LND and the approach are discussed. A special focus will be made on segmentectomy in Section 3.2.2.

#### 3.2.2. “Resecting Less”, Segmentectomy for Early-Stage NSCLC, from Present to Future

Despite resecting less, segmentectomy must lead to a complete resection of the tumor with safe margins and LND providing accurate staging and preserving long-term outcomes. Segmentectomy is indicated not only for compromised patients [84], but also for specific cases and selected patients [28,29,85], with a pure ground-glass opacity tumor of <2 cm, or an adenocarcinoma in situ of <2 cm, or a minimally invasive adenocarcinoma or an invasive adenocarcinoma of <2 cm [86] identified on screening CT scan, if expected margins are >1 cm or measuring at least the size of the tumor. Concerning short-term outcomes, compared to lobectomy, results are debated and segmentectomy does not seem to improve them that much [87,88] due to air leak, which is the main post-operative complication.

The technical challenge of segmentectomy is to perform a true anatomical and oncologic resection with safe margins. A multimodality imaging approach [89,90] to segmentectomy, using a combination of colored or fluoroscopic endoscopic [91,92,93] or physical X-ray [94,95] markings and 3D reconstructions of segmental vessels and bronchus, segments and tumor [89,96,97], helps surgeons to identify small, non-visible, and non-palpable tumors, and to anticipate individual anatomy in the surgical space and expected oncologic margins to perform oncologically effective and safe personalized tailored segmentectomy (Figure 4). These imaging tools are helpful for open segmentectomy, and even more for minimally invasive approaches, using indocyanine green and near-infrared angiography [98,99,100] to identify the intersegment plan.

#### 3.2.3. Surgical Approaches for Segmentectomy

Thoracotomy is the main approach for segmentectomy as for lobectomy but today, VATS and RATS approaches are recognised as safe and effective for this indication, providing the best short-term outcomes [101,102,103,104] and good long-term outcomes [101,105,106] compared to thoracotomy. Nevertheless, the superiority of VATS or RATS is still debated with conflicting results [107,108,109,110]. VATS segmentectomy is challenging compared to an open approach, and RATS allowing us to “mimic an open procedure” is therefore an interesting approach compared to VATS. Nevertheless, due to logistic and economic reasons, VATS remains a validated approach for “easier segmentectomy” [87]. Finally, it is important to remember that thoracotomy, VATS and RATS approaches for segmentectomy are complementary and should no longer be opposed.

### 3.3. Surgical Options for Resectable Stage II NSCLC Excluding cT3 N0

An essential prerequisite is pre-operative multimodality imaging and multimodality lymph node assessment [111], including ultrasonography guided fine needle biopsy and surgical exploration, which are essential in multidisciplinary discussion to define the best personalized treatment and the best outcomes for patients [112].

For stages IIA and B NSCLC (Figure 5), including T2b N0 or T1 to T2b N1 but excluding T3 N0, surgery is recommended, if possible, defined as an anatomical and oncologic resection as lobectomy, bilobectomy or pneumonectomy with complete LND by thoracotomy. Minimally invasive approaches are not strictly contraindicated, but are reserved for selected patients in experienced centers according to recommendations [28,29,85,113]. It is important to note that the approach only represents the means to perform an oncologic and anatomical “RO resection”. Today, thoracotomy allows excellent post-operative outcomes thanks to the modern management of patients. Recent authors have reported better short-term outcomes for VATS and RATS, with the same long-term outcomes [53,54,55] compared to thoracotomy but this implies a mastery of VATS and RATS approaches for these more complex cases.

### 3.4. Surgical Options for Stage III NSCLC Including T3, from Present to Future

Stage III NSCLC groups together multiple tumor presentations from T1 to T4 and N0 to N3 (Figure 6). cT3 N0 NSCLC are presented together here due to their “similar management”.

#### 3.4.1. Surgery for cT3 NSCLC

cT3 tumors group together multiple presentations of NSCLC, including high volume tumors between 5 and 7 cm, multiple tumor nodules in the same lobe or tumors infiltrating the chest wall including Pancoast-Tobias tumors, and tumors infiltrating the phrenic nerve, or the pericardium (Figure 6). 

For cT3 NSCLC with additional tumor nodules in the same lobe or a high volume tumor between 5 and 7 cm, N0 or N1, non-N2 involved, the gold standard lung resection is lobectomy with complete LND by thoracotomy [64,113]. Nevertheless, a minimally invasive approach could be used if an “R0 resection” can be done. In case of single site N2 involvement, neo-adjuvant chemotherapy could be proposed. After chemotherapy, open lobectomy with complete LND is recommended [64,113], but minimally invasive approaches could be performed by expert teams. 

For cT3 NSCLC tumors with chest wall invasion, guidelines are precise for Pancoast-Tobias tumors, recommending multimodality treatment combining *en bloc* lobectomy with the involved chest wall structures and LND [64,113,114] after neo-adjuvant cisplatin-based chemotherapy associated with radiotherapy. The main goal of surgical resection is to achieve complete resection with safe margins. Two surgical approaches have been described [114]: the historical posterior thoracotomy—the hook modification of the Shaw-Paulson incision—and the anterior approach—popularized by Dartevelle. For some specific cases, and selected patients, a hybrid approach could be performed, combining a VATS or a RATS approach and thoracotomy, or alternatively a totally minimally invasive approach by VATS or RATS, allowing complete resection with safe margins in expert surgical teams [115,116,117] and providing excellent short-term outcomes. Nevertheless, a totally minimally invasive approach is for highly selected patients and expert surgical teams. A prior VATS exploration can always be performed in order to avoid thoracotomy in case of unexpected pleural involvement.

For cT3 NSCLC tumors, with non-Pancoast-Tobias chest wall involvement, N0/N1 or single site involved N2, M0 NSCLC, neoadjuvant therapy is less defined and optimal treatment options are still debated. Neo-adjuvant chemotherapy is considered in case of node involvement, but parietal radiotherapy is discussed for each case and is indicated if safe surgical margins cannot be obtained. Nevertheless, a resectable tumor should be managed in order to achieve complete resection [64,113]. An open approach is recommended, but a prior VATS exploration can always be done to avoid thoracotomy in case of unexpected pleural involvement, and for selected patients a minimally invasive anatomical resection could be done followed then by elective thoracotomy for *en bloc* resection. 

#### 3.4.2. Surgery for cT4 NSCLC

cT4 tumors group together multiple presentations of NSCLC, including high volume tumors greater than 7 cm, separate tumor nodule(s) in other ipsilateral lobes, a central and hilar tumor, or a tumor infiltrating the carina, the trachea, great vessels, the heart, the mediastinum, the vertebral body, the esophagus, the phrenic nerve, the laryngeal recurrent nerve, and each treatment needs to be personalized. A special focus on pneumonectomy will be made in Section 3.4.4 and will not be discussed here.

For resectable central and hilar T4 N0/N1 M0 NSCLC, and cT4 NSCLC tumors invading other structures, resection is recommended in a specialized center [64,113]. Neo-adjuvant chemoradiotherapy is also less defined compared to Pancoast-Tobias treatment, and is considered as a second option therapy for tumors that could be resectable [64,113,118].

For ipsilateral cT4 NSCLC lung nodules, anatomical resection is recommended [64,113,118] and segmentectomy could be done in different lobes or associated with lobectomy according to patients’ functional status.

#### 3.4.3. Redefining the Role of Pneumonectomy, “Surgical Haute Couture”

For central and hilar NSCLC, pneumonectomy could be indicated. Sleeve lobectomy was initially performed for patients with a contraindication to pneumonectomy. And as an echo of the “shift from pneumonectomy to lobectomy in the 50s”, anatomical sleeve resection is a validated oncologic alternative not only to prevent the risk of residual tumor, but also to reduce post-operative mortality and to prevent a reduced QoL due to pneumonectomy [119,120,121,122,123]. Thoracotomy is the main approach for sleeve resection, but there is no absolute contraindication to using a minimally invasive approach as VATS or RATS, representing a validated oncologic and functional alternative to open sleeve lobectomy or pneumonectomy without compromising oncologic prognosis [124,125,126,127,128]. In a modern surgical era, these patients should also be proposed preoperative rehabilitation and ERAS protocols in order to reduce postoperative complications and to enhance recovery after surgery.

#### 3.4.4. Locally Advanced NSCLC

For T1, T2a and T2b, N1 or with single-site N2 involvement, MMT combining chemotherapy and/or radiotherapy followed by lung tumor resection improves survival, with good short- and long-term outcomes [129,130,131,132,133,134]. Prognosis factors associated with improved OS are T and N downstaging, pathological complete response (pCR) [129,130,131,134], and non-pneumonectomy resection. Nevertheless, few patients achieved pCR after a regimen of neo-adjuvant chemotherapy [135,136]. For patients with pCR on surgical specimen, after neo-adjuvant therapy, clinical stage did not impact survival, with a similar 5-year OS of 66.1%, 60.9% and 58.6% (*p* = 0.28), respectively, for clinical stages I, II and III [129]. This underlines the need to precisely assess the NSCLC both before and after neo-adjuvant treatment in order not to exclude patients with stage III tumors from surgery. 

#### 3.4.5. Surgery for N3 NSCLC

At the margin of surgery, because for N3 NSCLC, surgery is generally not recommended and remains an exception [29,113], but it could be discussed for some highly selected patients [137], with improved long-term survival in favor of an MMT plan.

### 3.5. Surgery for Stage IV NSCLC

Surgery is currently indicated for oligometastatic NSCLC, but thanks to advances in targeted therapy and immunotherapy [82,138] surgical indications for NSCLC metastatic disease have been redefined in an MMT plan.

For M1a contralateral lobe tumor, non-N2, with non-brain/adrenal metastasis, patients can be treated by lung resection, respecting anatomical dissection and LND, if the patient has adequate pulmonary function [64].

For isolated brain and adrenal metastasis, non-N2, with no other site of metastasis, the resection of the primary lung tumor and the brain or adrenal metastasis is recommended with the longest survival observed, compared to treatment excluding local therapy [64,138,139,140,141]. 

Local therapies, including ablative options as surgery and non-ablative treatment as radiotherapy and radiofrequency for example, must be discussed compared to maintenance for patients without disease progression after initial systemic therapy [140]. Moreover, for patients with stage IV NSCLC initially controlled by systemic therapy, but newly progressing in one or two metastatic sites, local therapy including surgery could spare systemic therapy, treating local progression by local options, and saving a new regimen of systemic therapy for “more severe disease progression”. Of course, these very specific cases must be discussed in a multidisciplinary meeting.

### 3.6. “Recuperative Surgery” and “Salvage Surgery”

“Recuperative surgery“ also known as “salvage surgery” is considered for patients with a residual local tumor or recurrence after medical treatments without surgical resection. Until recently, the treatment used was chemotherapy and radiotherapy, but today, patients are treated with a combination of chemotherapy, targeted therapy and immunotherapy and different regimens of radiotherapy including SRBT. “This surgery” is not a new indication but has evolved due to innovative treatments with new surgical challenges [142,143,144,145,146,147,148]. Nevertheless, in recent literature, including selected patients, because indications are rare and patients must be selected, “salvage surgery” as “recuperative surgery” could be considered with controlled morbimortality and long-term outcomes. For these indications, and selected cases, minimally invasive procedures, alone or as part of a “hybrid approach”, are being developed to reduce morbidity and mortality and to preserve QoL while preserving long-term survival [145,147,148].

## 4. Surgery in the Diagnosis and Treatment of Symptoms

When curing cancer is not possible, surgeons can have a symptomatic or a diagnostic role. 

For palliative patients, with dyspnea due to malignant pleural effusion, surgery can be beneficial thanks to talc poudrage and talc slurry in order to relieve patients [149]. Indwelling pleural catheter is an alternative approach for trapped lung with inferior definitive pleurodesis but with a comparable control of breathlessness. For these patients, QoL, breathlessness, oncologic outcomes and patient preferences are essential to inform clinical decision making. 

Recurrence is part of the natural history of lung cancer. Today, patients are more frequently followed-up and recurrences can be detected earlier, before symptoms occur. Recurrences can be divided into 2 groups according to whether surgical curative resection can be performed or not. 

If curative resection cannot be performed, the role of surgery is to provide a precise diagnosis based on histology and biomolecular characteristics in order to propose systemic innovative personalized treatment. A combination of perioperative care, ERAS protocols, and minimally invasive surgery could aid diagnosis, allowing low morbidity and mortality, and improved patient comfort.

If curative lung resection can be performed, surgery has many advantages compared to non-ablative or less ablative treatment, as radiotherapy and radiofrequency for example, allowing histology and lymph node stage, with very low morbidity and short hospital stay or ambulatory surgical care [62,63,150,151] with improved recovery and survival. Nevertheless, lung resection needs to be tailored to patients’ functional characteristics and lung cancer progression, and a good alternative would be to combine surgical ablative and non-surgical ablative treatment to spare lung tissue, lung function, and to enhance recovery and survival.

## 5. Future Perspectives

### 5.1. From ERAS to Ambulatory Surgical Care

As an illustration of modern surgery with enhanced recovery, combining ERAS protocols, minimally invasive surgery, and modern anesthetic protocols, we will cite reports dealing with “fast track” major lung resection [61,62] or lung wedges [63] where patients are discharged at day one after the surgery [61]. These cases of “extreme patient management” illustrate the low functional impact of modern lung resection. Today, ambulatory-care thoracic surgery, or fast-track thoracic surgery is less and less considered as “extreme”, because patients’ parameters are controlled and patients are educated and well prepared. Fast-track thoracic surgery leads to the treatment of a suspected or confirmed NSCLC in a shorter time than radiotherapy, but with a definitive pathological analysis.

### 5.2. Redefining Functional Assessments

Another question raised for these patients is how we can estimate the operative risks of modern and minimally invasive surgery. Current guidelines [152] are mainly based on patients operated before the era of ERAS and minimally invasive surgery. Thus, there is an urgent need for modern functional assessment guidelines [153] to better assess the risk of post-operative complications, to better screen high-risk patients, and to better choose treatment. Modern lung cancer resection allows lower complication rates and better long-term outcomes compared to non-ablative treatment and should be proposed to patients according to modern guidelines [80]. 

### 5.3. Hybrid Surgical Approach or Local Treatment, “Thinking Green Surgery”

Like hybrid vehicles, in modern NSCLC management, we need to review treatment and approaches and consider using a combination of surgical/non-surgical treatment and surgical approaches for diagnosis and treatment. 

Imaging-guided surgery is increasingly used. Many tools are available to perform imaging-guided resection in order to anticipate patients’ anatomic variabilities, to guide precise resection as segmentectomy, or for training in order to test surgical devices for a complex case before the actual surgery [154]. These devices and tools could be used together to perform multimodality imaging-guided surgery. Moreover, for training, we can hypothesize that they could be integrated in the surgical procedure by a training session on a VATS or RATS simulator. The aim of these tools and devices is to reduce surgical risks and post-operative complications and also to enhance and improve surgical procedures as in aeronautics. VATS and RATS, using imaging technology, could easily integrate these devices and virtual reality in order to guide resection [89,96] for segmentectomy or wedge resections for metastasis. 

For major lung tumor, as T4 NSCLC invading the chest wall, or Pancoast-Tobias tumor or central hilar tumor, when resection can be performed, it would be useful to integrate all available tools, elective open procedures, VATS, and RATS and to manage the surgical procedure in a “damage-control way” to enhance recovery. The purpose is not “a strict application of minimally invasive surgery” but to integrate all approaches, as wider and more precise vision with modern optical cameras, for VATS and RATS, which is helpful for adhesiolysis, and combining anterior or posterior guided thoracotomy for chest wall resection with an intra-pleural view by VATS and RATS. These procedures could be associated with elective open access for specific dissection [145], for example, a subclavicular incision to dissect the subclavicular artery and vein from an apical tumor, before minimally invasive lobectomy, or spine dissection by elective open incision, for posterior chest wall resection before minimally invasive lobectomy, etc. Managing a surgical procedure with the same philosophy used in “damage-control surgery” means combining two distinct surgical procedures within 24 or 48 h, in order to reduce the time of the procedure, and to reduce post-operative stress. Between the two procedures, functional rehabilitation continues, with early feeding, physical activity with pain control, in order to enhance final recovery. 

Lung segmentectomy could be interesting in order to spare lung tissue without compromising long-term survival for suspected or diagnosed NSCLC. For non-operated patients, non-ablative treatment, as cryoablation, radiofrequency ablation, microwave ablation, stereotaxic body radiation, percutaneous ablation, are proposed as an interesting therapy [80,81,155,156]. Rather than comparing ablative surgery and non-ablative treatment, surgery and treatment could be combined with the objective of sparing lung tissue in patients with T4 or M1a lung nodules, who could be diagnosed with other NSCLC. 

### 5.4. Resecting after Innovative Systemic Treatment, Future Perspectives

Most lung cancers, approximately 70%, are diagnosed at an advanced stage [3], and could require a multimodality treatment. After a regimen of neo-adjuvant treatment, the main prognosis factors associated with improved OS are T and N downstaging, and pathological complete response (pCR) [129,130,131,134]. The pCR is strongly correlated with improved survivals, and reflects the neo-adjuvant therapy effect [129,130,131,134]. Recently, innovative systemic therapies were used in neo-adjuvant protocol in order to increase survivals, based on the hypothesis that these treatments would improve the pathological response. Very interesting results were observed with neo-adjuvant immunotherapy or targeted therapies associated with chemotherapy. 

Recently Allaeys et al. [157] have published a review reporting the latest data from trials and ongoing trials, concerning the short- and long-term postoperative follow-up of patients undergoing surgical resection after induction immunotherapy and/or targeted therapy with or without chemotherapy. In this review [157], surgery for a resectable locally advanced stage NSCLC after induction therapy including immunotherapy or targeted therapy with or without chemotherapy seems to be feasible and safe. Nevertheless, the lung resection could be more difficult with serious concerns due to hilar, vascular and bronchial fibrosis, but acceptable morbidity and mortality rates were reported in experienced centers. Concerning neo-adjuvant immunotherapy, in the first trial [158], major pathological response was reported in 45% patients from the immunotherapy group *vs* 10% without immunotherapy. Five adverse events, with one grade 3 were reported and the surgery was performed without the need for further delay. In other trials, higher major pathological response was reported, from 57% of major pathological response with 33% of pCR [159] and recently 83% of major pathological response with 63% of pCR [160]. These pCR results are far superior to previous neo-adjuvant protocols [135] with controlled adverse events allowing better survivals, with 77% of 2-year DFS, and 90% of 2-year OS for stage IIA NSCLC. As reported by Allaeys et al. [157] many trials reported acceptable safety and efficacy of neo-adjuvant immunotherapy with controlled adverse events and post-operative morbidity and a surgery performed on time without the need to postpone it. Bott et al. [161] reported a higher rate of conversion to thoracotomy due to significant hilar inflammation and fibrosis which are the main surgical technical difficulty reported both in minimally invasive approach or for open procedures. In the IFCT-1601 IONESCO trial [162], due to a high-rate of 90-day mortality, 9%, this trial should be stopped early, nevertheless this high mortality was most likely related to significant patients’ comorbidities than durvalumab itself. 

Concerning neo-adjuvant targeted therapies, it seems to be a safe and effective treatment allowing major pathological response and pCR with controlled adverse events, and post-operative morbidity and mortality rates based on the reviews from Allaeys et al. [157] and Li Sun et al. [163]. Nevertheless, phase III trials are necessary to confirm these findings.

Neo-adjuvant immunotherapy and targeted therapies with or without chemotherapy lead to a shift in NSCLC treatments, and in the future, promising results with validated long-term survivals, OS and DFS are expected. 

Maybe, with this emerging literature on this subject, and these impressive results, we could consider surgery as an adjuvant treatment and not only as a “salvage strategy” for patients priorly treated by a full regimen of chemotherapy with targeted therapy or immunotherapy [164,165,166]. These rare cases need to be discussed in multidisciplinary meeting with expert teams. 

### 5.5. Providing Adequate Cares for All

Lung cancer screening programs, physiotherapy, smoking cessation, comorbidities’ treatments, ERAS protocols, surgery, and especially RATS, medical treatments and innovative systemic therapies are cost effective. In this review, we have underlined past, present and future roles of thoracic surgery, in high-income country, with an advanced health care system. “But the reality is quite different”. About 60% of cancers occur in a disadvantaged population [167], and concerning lung cancer, incidence and mortality rates have increased in many low- and middle-income countries [1,2,168]. So, our review may not cover the reality of lung cancer management around the world.

Even in high-income countries, patients with a lower level of income and education are less likely to participate in preventative health checks as cancer screening. However, due to more bad lifestyle habits in this disadvantaged population, a higher incidence of lung cancer is found [169]. In addition, the cost of surgical treatment varies depending on the patient’s demographic background and medical history, as well as the costs associated with the occurrence of adverse post-operative events. Thus, treating a disadvantaged population is more expensive which adds another disadvantage for this population [170]. Developing a robotic thoracic program is also associated with higher hospitals costs compared to VATS and open approaches [109,171,172,173,174]. Nevertheless, these additional costs eventually are smoothed out with the use of the robot and are no longer significant [173]. Nevertheless, an ERAS program and a VATS minimally invasive surgery, lead to an overall reduction in the costs of surgical treatments, by reducing adverse post-operative events, complications and reducing hospital length of stay [38,175,176]. 

Today, for still providing better care by improving the survival and quality of life of our patients, each practitioner has a duty of good economic management of the resources he or she uses. Not to just save money, but to allow more resources for prevention programs for populations at risk, for example.

In low- and middle-income countries, lung cancer incidence and mortality rates have increased due to increases in smoking tobacco, or water piper or cannabis smoking, toxic exposure as asbestos, dust, fumes, nickel, silica and insecticides [167]. In these countries, health care systems, cannot always carry out prevention, information, education policies, and investment in hospital medical means, and in innovative treatments [177]. But as reported before, some innovative peri-operative treatments as ERAS protocol and VATS make it possible to minimize health care costs. In order to provide the best care, specific guidelines based on real-life management of patients from specific region [178] are needed.

In every country, whether low or high income, there are significant medical needs that require new approaches to optimize patient outcomes and quality of life, which remains one of the greatest challenges of modern medicine. 

## 6. Conclusions

Surgery remains the cornerstone treatment for early-stage NSCLC and could be an integral part of an MMT plan for advanced-stage cancer associated with innovative therapies. Lobectomy with LND is the gold standard resection but needs to be adapted to lung cancer characteristics and patient lung function. Surgeons must do more than just performing the best oncologic, personalized and tailored resection, with the best short- and long-term outcomes, they must also promote good health, and thus have a determining role in providing patient information and education to promote cohesion in ERAS programs. Minimally invasive surgery, with VATS and RATS and modern perioperative anesthetic management lead to enhanced patient recovery with the best short-term outcomes and good long-term survival. VATS and RATS are indicated for early-stage NSCLC. However, more advanced stages require expert surgical teams. Moreover, because lung cancer is a chronic disease, surgeons also contribute to patient follow-up in order to detect early recurrences. Sparing lung tissue, preserving lung function and allowing longer survival are goals to be achieved considering lung cancer as a chronic disease. To reach these objectives, segmentectomy and sleeve resection are validated oncologic resections that provide long-term survival and good short-term outcomes.

Beyond the opposition of surgical approaches, a hybrid approach, combining open, VATS and RATS, and surgical ablative and non-ablative treatments, for complex cases, could be used, in a “damage-control-like” MMT plan in order to perform the best oncologic resection, with improved short-term outcomes and preserved long-term survival. 

Lung cancer has evolved and surgeons have evolved their practices like other physicians. Through a standardized procedure, surgeons continue to propose personalized resection, tailored for each patient, with better outcomes than yesterday, and with even better outcomes tomorrow.

## Figures and Tables

**Figure 1 cancers-13-03711-f001:**
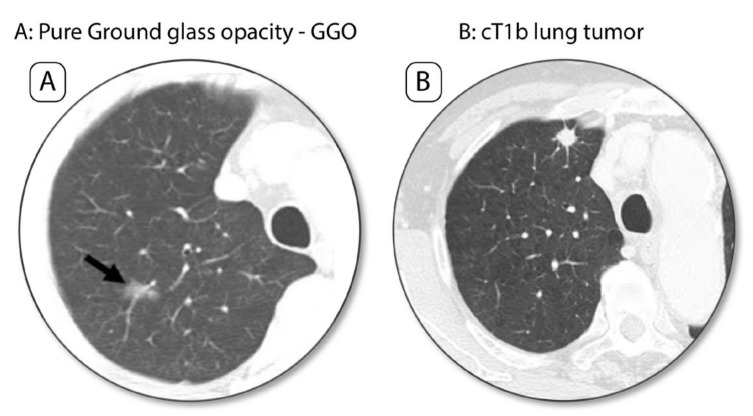
Early-stage lung cancer on screening CT scan. (**A**) Pure Ground glass-opacity (GGO) in the right upper lobe. (**B**) Lung cancer nodule, cT1b as a densified lung nodule.

**Figure 2 cancers-13-03711-f002:**
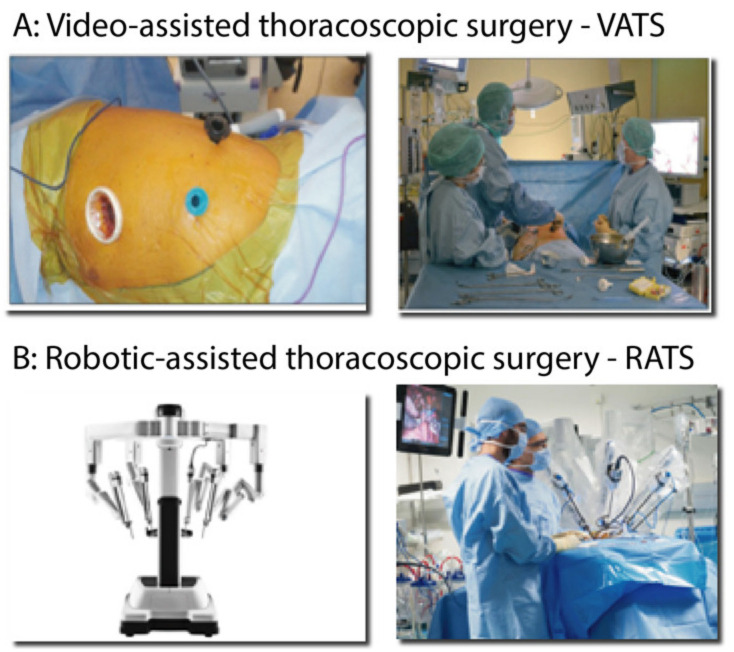
Video-assisted and Robotic-assisted thoracoscopic surgeries. (**A**) Surgical skin incisions for a left lung resection by video-assisted thoracoscopic approach (VATS) and operators setting in the operating room. (**B**) da Vinci (Intuitive Surgical) Xi platform for robotic-assisted thoracoscopic surgery (RATS) and assistant surgeons near the patient during left lung surgery performed with the X platform.

**Figure 3 cancers-13-03711-f003:**
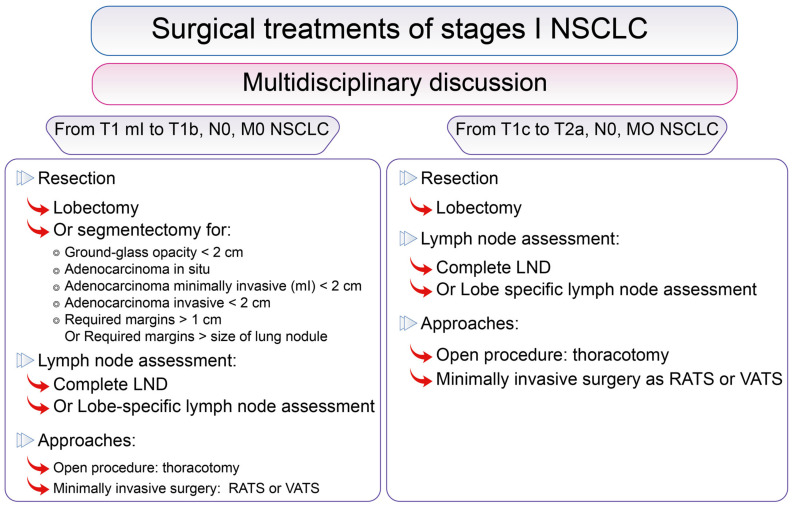
Surgical options and treatments for resectable Stage I lung cancer. Surgical options and treatments according to lung cancer stage in the lung cancer Stage I group according to the 8th Edition of the TNM classification of lung cancers [83].

**Figure 4 cancers-13-03711-f004:**
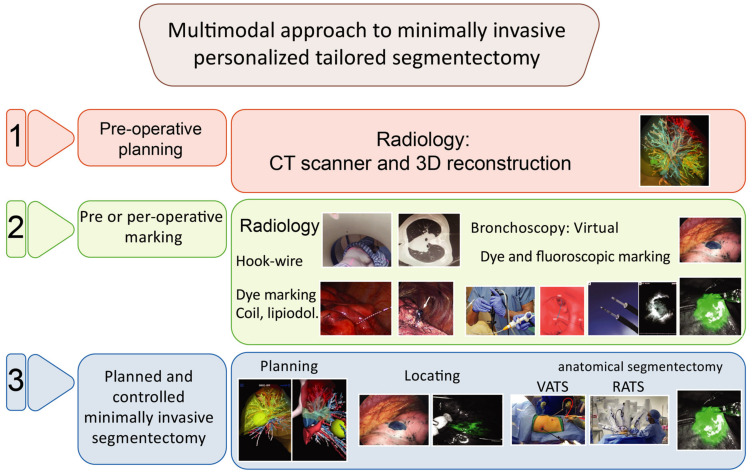
Multimodal approach to minimally invasive personalized tailored segmentectomy. Multimodal approach to minimally invasive personalized tailored segmentectomy in 3 steps. **Step 1**: Pre-operative imaging with CT scan and 3D reconstructions in order to identify anatomical variabilities to perform anatomical segmentectomy. **Step 2**: Pre or per-operative lung tumor marking by radiological technique combining CT scan mapping, and hook wire, dye marking, lipiodol or coil; or bronchoscopy technique with virtual endoscopy and ultra-thin probes allowing dye marking or fluoroscopic marking with indocyanine green. **Step 3**: Planned and controlled minimally invasive segmentectomy with planning validation, guided resection and the identification or the intersegmental plan using fluoroscopic marking during a VATS or a RATS approach.

**Figure 5 cancers-13-03711-f005:**
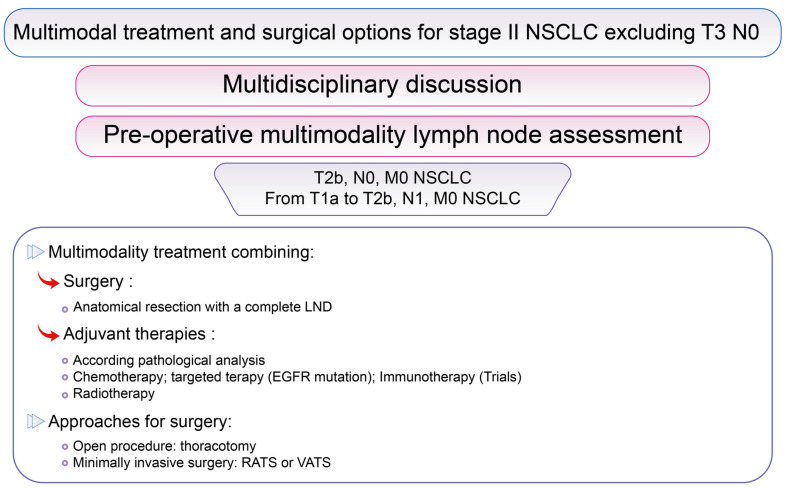
Multimodal treatment and surgical options for stage II NSCLC excluding T3 N0. Surgical options according to lung cancer stage in the lung cancer Stage II group excluding T3 N0, according to the 8th Edition of the TNM classification of Lung Cancer [83].

**Figure 6 cancers-13-03711-f006:**
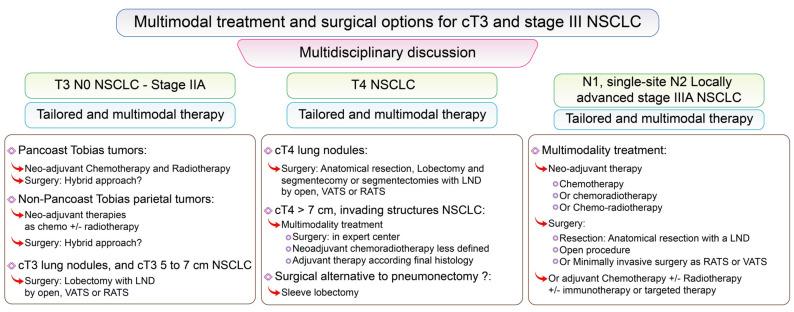
Multimodal treatment and surgical options for cT3 and stage III NSCLC. Surgical options according to lung cancer stage in the lung cancer Stage III group and cT3, according to the 8th Edition of the TNM Classification of Lung Cancer [83].

## Data Availability

The data presented in this study are openly available. Each results of studies, or reports are referenced by a reference number and the complete reference is available at the end of the manuscript, with authors, journal ref, year.

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
