# Peer review of "The Role of Surgery in Lung Cancer Treatment: Present Indications and Future Perspectives—State of the Art"

_cancers, 2021, doi:10.3390/cancers13153711_

Round 1

Reviewer 1 Report

The authors provide a comprehensive review of the contemporary role of thoracic surgery in the diagnosis and management of the different lung cancer stages. Most recent evolutions are discussed including enhanced recovery programs. The authors also pay attention to a more generalized, integrated approach with emphasis on the necessity of changing the lifestyle of patients with lung cancer.  

Comments:

- fig. 1 legend B Lung cancer node: in fact this is a solid nodule; “node” refers to “lymph node” which is quite different; this should be corrected throughout the manuscript as this is several times mentioned; in this way, any confusion is avoided

- fig. 3 left part: replace “expected margins > 1cm or size of tumour” by “margins > 1cm required or margins > size of tumour” which I suppose, the authors want to indicate

- fig. 5 1st line “Multimodality” instead of “Multimodali”

- fig. 5 “Or segmentectomy for” seems to be copied from fig. 3; for stage II at least a lobectomy is required

- fig. 6: cT3 lung nodes and cT4 lung nodes: see previous comment; just indicate cT3 and cT4

- fig. 6 right part: Neo-adjuvant chemotherapy: add “or chemoradiotherapy or chemo-immunotherapy”

- induction treatment comprising immunotherapy has recently been shown to significantly increase response rate compared to chemotherapy only, especially for N2 disease. This should be discussed by the authors. The NADIM trial has been fully published and should be included [Provencio M et al. Neoadjuvant chemotherapy and nivolumab in resectable non-small-cell lung cancer (NADIM): an open-label, multicentre, single-arm, phase 2 trial. Lancet Oncol . 2020 Nov;21(11):1413-1422]. Many other trials are currently ongoing.

Author Response

Dear reviewer,

Best reagards

François Montagne, Florian Guisier, Nicolas Venissac, Jean-Marc Baste

Reviewer 2 Report

This is a comprehensive review of the current surgical therapy in managing lung cancer patients. The authors have covered all the important aspects and should be commended to put all this information together.

I have a few minor suggestions:

(1) It is worthwhile to add a paragraph mentioning that the VATS technology has been evolving dramatically within recent decades. The single/uniportal VATS technique developed in 2011 has been widely adopted and should be considered as a promising major advance in lung surgery.

(2) It is also important to recognize that not all countries in the world have been benefitted from the advance of lung cancer surgery. It is still a challenge to expand the minimally invasive surgery techniques to low-income countries. Even within one country, RATS may only be available in large tertiary centers. The cost-effectiveness of RATS should also be evaluated in different healthcare systems.

(3) Since the size of the incision is one of the major concerns for both RATS and VATS, you may considering cite the following paper as it addresses a major challenge:
Sihoe ADL et al. Technique for delivering large tumors in video-assisted thoracoscopic lobectomy. Asian Cardiovasc Thorac Ann 2014;22(3):319-28

Author Response

Dear reviewer,

Please see the attachment.
Best reagards

François Montagne, Florian Guisier, Nicolas Venissac, Jean-Marc Baste

Reviewer 3 Report

Interesting review on the field discussing the evolution on lung cancer treatment and the present role of surgery.

Author Response

The authors thank reviewer #3 for his work in proofreading and assessing this review.

Round 2

Reviewer 3 Report

It's an interesting review, but not a paper reporting a new, personal scientific experience